# The Acute Effects of Schoolbag Loading on Posture and Gait Mechanics in 10- to 13-Year-Old Children: A Cohort from the North West Province

**DOI:** 10.3390/children10091497

**Published:** 2023-09-01

**Authors:** Bridget Grobler, Mark Kramer

**Affiliations:** Physical Activity, Sport, and Recreation (PhASRec) Research Focus Area, North-West University, Potchefstroom 2531, South Africa; groblerkatryn@gmail.com

**Keywords:** adolescent, backpack, kinetics, kinematics, South Africa

## Abstract

Most schoolchildren carry schoolbags, of which a substantial proportion carry loads that exceed 15% of their body mass. Although the effects of loading have been investigated to varying degrees, the status of schoolbag loading and the acute affects thereof on gait and posture have not been thoroughly investigated within the South African context. A total of 60 participants in the 10–13-year age range volunteered for the present study. Significant differences were evident for relative load carriage (χ^2^(3) = 14.54, *p* < 0.001), forefoot and heel forces (M_diff_ = 17.05–34.86 N, *p* < 0.001), force ratios (M_diff_ = 0.02, *p* = 0.029), and gait speed (M_diff_ = −0.18 km/hr, *p* = 0.016), but not for any postural angles (M_diff_ = −3.37–6.08 deg, all *p* > 0.052). Those who exceeded 15% BM were ~9 times more likely to report pain than those below 15% BM. The children in the current study carried significantly heavier relative loads (*p* < 0.001) compared to similarly aged children from other countries. Loading leads to acute changes in posture and gait that are likely not meaningful. However, excessive loading (>15% BM) leads to significantly higher perceptions and reporting of pain in 10–13-year-old children.

## 1. Introduction

Schoolbag carriage is a necessary component for most schoolchildren that, when excessive, can (i) compromise posture, (ii) induce neuro-musculoskeletal disorders, (iii) moderate cardiopulmonary function, and (iv) cause shoulder, back, and hand pain. The primary mediating factor for adverse reactions associated with schoolbag carriage is likely related to the actual mass of the schoolbag and, because of the nature of carrying, places most of the mechanical stress on the cervical vertebrae, shoulders, and lower back [1,2,3,4,5]. Heavy schoolbag loads are also typically associated with postural deviations such as posterior pelvic tilt [6], increased thoracic kyphosis [7], altered postural angles (e.g., cranio-vertebral angle [CVA], cranio-horizontal angle [CHA], sagittal shoulder posture [SSP], and lumbo-sacral angle) [8,9,10,11,12], increased forward trunk lean [13,14,15], changes in lung function [1,16], and changes in both gait kinetics and kinematics [17,18,19]. Such changes appear to manifest when the load exceeds 10–15% of the body mass (BM) of the child [8,9], with the prevalence of self-reported lower back pain in children aged 8 to 13 years old being ~32% [20]. Although substantial changes of 4–9° have been recorded in key upper body postural angles with *acute* schoolbag loading of ~18% BM, there appears to be substantial variability in both magnitude and direction of these changes as a function of loading [10,12,13,21,22,23]. Whether or not these changes were associated with increased perceptions of pain or longer-term postural alterations remains an under-studied component of load-related research, especially within the African context [24].

Although research on the effects of schoolbag mass on posture and gait has been conducted internationally, to date only one study has been conducted within the South African context [24,25]. Importantly, the latter study only recorded body mass and bag mass and used a questionnaire to infer aspects related to bag carriage and pain. Therefore, it is unclear whether the mass of the schoolbag has changed over the past two decades, what effects this mass has on key postural angles and gait parameters, and whether pain related to load carriage is experienced by individuals across different age ranges. More specifically, children in the 10–13 age range are considered particularly vulnerable on the basis that younger age cohorts, partly because of growth spurts, may experience a greater possibility for maladaptive responses of soft tissue structures when subjected to excessive loading [26,27]. For example, Neuschwander et al. [26] showed that schoolbag loads ranging from 10–30% of BM led to substantial reductions in vertebral disc heights and greater lumbar asymmetries in children, which may at least partially explain some of the pain experienced in such age cohorts. Similar findings were established for load-related changes in the curvatures of the cervical and lumbar regions of the spine, which in turn may lead to muscle imbalances and compensatory changes in body posture during static and dynamic tasks [4,28,29].

South Africa is classified as a developing country and, more importantly, has a high cost, low-performance education system that is marred by poor management of funds and high failure rates [30]. It is therefore not surprising to find that the educational standards within South Africa are poor compared to similarly developing countries (e.g., Brazil, Ethiopia, or India), ranking 127th out of 157 countries based on data from the World Bank [31,32,33]. Whether such issues manifest at the school level by affecting access to textbooks and lockers, and the outcomes of this on the subsequent loads carried by children has not been previously investigated, especially in comparison to other countries.

In this context, the objectives of the present study were fivefold, namely, to investigate whether (i) the relative mass of the bag carried by children changed as a function of age, (ii) the load carriage led to significant changes in postural angles, (iii) the load carriage significantly altered gait-related parameters (e.g., forefoot force, heel force, and gait speed), (iv) differences in perceived pain were evident in those carrying heavier loads (i.e., >15% BM), and (v) the relative loads carried in the present study were significantly different from similarly aged children from other countries. We hypothesized that load carriage would lead to significant static and dynamic differences in school-going children, but that the relative loads carried would be similar across countries.

## 2. Materials and Methods

### 2.1. Participants

A total of 60 participants volunteered for the study and were subcategorized by age group, namely, 10-year-olds (*n* = 15 (Male = 8; Female = 7); height: 1.39 ± 0.14 m; weight: 34.03 ± 9.93 kg), 11-year-olds (*n* = 15 (Male = 10; Female = 5); height: 1.40 ± 0.08 m; weight: 32.98 ± 7.91 kg), 12-year-olds (*n* = 16 (Male = 12; Female = 4); height: 1.58 ± 0.08 m; weight: 43.94 ± 5.77 kg), and 13-year-olds (*n* = 14 (Male = 7; Female = 7); height: 1.55 ± 0.09 m; weight: 47.57 ± 15.26 kg). Age classification within the 10-year-old cohort would imply that a child is aged between 10.0 and 10.99 years; this is true for all other age classification cohorts. A sample size of 25–71 participants was calculated based on a mean effect size of d_z_ = 0.30–0.52 for differences in forefoot and heel pressures, as well as postural angles during loaded and unloaded conditions in similarly aged cohorts [4,13,29]. Inclusion criteria consisted of (i) those aged between 10 and 13 years, (ii) only those with double-strap bags, (iii) those free from injuries that might prohibit walking or conventional carrying of the schoolbag, and (iv) those who submit completed informed assent and consent forms. Exclusion criteria consisted of (i) any recent systemic illness, (ii) musculoskeletal, cardiorespiratory, and/or neurological problems that would hinder the ability to complete the tasks pertinent to the study, and (iii) any congenital deformities.

### 2.2. Procedures

Participants entered the designated test area and were ushered to the first testing station, which included recording of anthropometric and schoolbag data such as body mass (kg), height (m), and schoolbag mass (kg). Subsequently, the participants were prepared for gait and posture analyses during both unloaded and loaded conditions. The postural analysis involved standing on the pressure platform as motionless as possible for a period of 20 s. A photograph (Sony Cyber-Shot DSC-RX10 MK III, Sony, New York, NY, USA) was taken in the sagittal plane at the 10 s mark. Photographs were extracted at a later stage and imported into specialist software for the derivation of key angles. Finally, participants were required to complete a series of steps across a pressure platform during both unloaded and loaded conditions, so that force and gait speed could be recorded for approximately 30 steps for each participant.

#### 2.2.1. Posture

To facilitate angle measurements, reflective markers were placed on the external canthus of the eye, tragus of the ear, spinous process of the 7th cervical vertebrae, and mid-point of the greater tuberosity of the humerus and posterior aspect of acromion (see Hande et al. [12] for details). Digital photographs were taken of each participant and loaded into specialized software for data extraction and analysis (IC Measure, version 2.0.0.161, The Imaging Source Europe GmbH, Bremen, Germany). The camera was placed 7 m from the location of the participant at approximate hip height, and the zoom function was used to adequately capture the pertinent markers for the angle measurements. The process was completed for both unloaded and loaded trials.

#### 2.2.2. Gait

The Zebris pressure platform (FDM-1.5, Zebris Medical GmbH, Weitnau, Germany) was used to record the force and pressure values during normal gait. Participants were instructed to walk a ~5 m length, which included the 1.5 m platform (i.e., 1.5 m before and after contacting the platform). A total of 3 steps would be recorded for each length, which was repeated for a total of 10 lengths (~30 steps). The procedure was replicated for the loaded condition. The average forefoot force (N), heel force (N), and gait speed (km/hr) across all recorded steps were retained for analysis for each participant.

#### 2.2.3. Backpack Questionnaire

To gauge specific characteristics related to the schoolbag, its transport, and the perceived pain that may be associated with transport, participants were asked to complete the backpack questionnaire [34]. Although the questionnaire included site-specific details of load-related pain (e.g., shoulder, neck, back, etc.), the present study investigated the over-arching presence of pain associated with loading, which was retained for analysis.

#### 2.2.4. Literature Comparisons

To contextualize the findings of the present study, we incorporated a quasi-meta-analytic approach whereby an extensive literature search was conducted that adhered to the guidelines set by the preferred reporting items for systematic reviews and meta-analyses (PRISMA) [35]. Articles were included that focused specifically on load carriage in children and were identified through databases such as CINHAL, LISTA, and PubMed. The selected keywords were as follows: children, adolescents, load carriage, schoolbag, backpack, pupils, gait, walking, lung function, kinematics, posture, and pressure. More specifically, studies were included if they met the following criteria:Full-length article published in English in a peer-reviewed journal;Experimental, quasi-experimental, observational, or cohort study design;Provided information related to the mass of the children and the bags carried.

Studies were excluded if:
They provided incomplete information such that relative load carriage could not be determined.

A total of 77 articles were initially identified. This number was reduced to 55 after the screening of abstracts and further reduced to 40 after application of the inclusion and exclusion criteria. Data from the same country were then pooled to provide overall unique averages for age and relative bag mass for each country, resulting in a final dataset consisting of 20 countries (see Appendix A).

### 2.3. Statistical Analyses

Data were evaluated for normality using the Shapiro–Wilk test, whereby deviations from normality were accepted at *p* < 0.05. All data are reported as mean ± SD unless otherwise stated. For the first objective, a nonparametric ANOVA (Kruskal–Wallis) was completed once all assumptions were checked (normality (Shapiro–Wilk), equality of variance (Levene)) to evaluate age-related differences in relative load carriage [36]. The post hoc Holm correction was used to adjust for multiple between-group comparisons. For the second and third objectives, the Wilcoxon signed-rank test was used to evaluate within-group differences caused by loading for each postural angle and gait parameter, respectively. A one-way ANOVA was used to assess differences in body mass between gender and age groups. To evaluate whether differences in reported pain were evident as a function of loading (i.e., > 15% BM) and gender, the Pearson chi-square test was used, whereby Cramer’s V served as the standardized effect size measure [37]. An unplanned analysis entailed an evaluation of the relationship between the relative mass of the bag and the change in postural angles caused by loading using a regression analysis. The Pearson correlation coefficient was used as a measure of association and interpreted in absolute terms as follows: negligible: 0.00–0.10, weak: 0.10–0.39, moderate: 0.40–0.69, strong: 0.70–0.89, and very strong: 0.90–1.00 [38]. Finally, we compared the relative bag mass of children in the present study to those from other countries using an independent two-sided *t*-test where the *p*-values were adjusted using a Holm correction for multiple comparisons. All statistical analyses were completed using R (RStudio (version 22.12.0 Build 353): Integrated Development for R. RStudio, PBC, Boston, MA, USA; URL: http://www.rstudio.com, accessed on 22 March 2023) [37,39].

## 3. Results

Data for body mass and bag mass for each age cohort are shown in Figure 1. There is considerable overlap in both body mass and bag mass across the different age cohorts, which is statistically evaluated in Figure 2.

No differences were evident for the main effect of body mass as a function of gender, regardless of age (F(1,52) = 1.54, *p* = 0.220), nor for the interaction effect between gender and age (F(3,52) = 0.59, *p* = 0.625). The differences in *relative* bag mass between age cohorts are shown in Figure 2. Relative bag mass remains relatively stable until age 12, whereafter a significant reduction is noticeable. More precisely, the relative bag mass is significantly lower in 13-year-olds compared to 12-year-olds (*p* < 0.001) and 10-year-olds (*p* = 0.020). Approximately 58% of the participants exceeded load carriages of 15% BM, and 27% of participants exceed load carriages of 20% BM.

The load-related differences in the postural angles are shown in Figure 3. The results indicate that loading does not lead to significant changes in any of the three key postural angles. It is important to highlight the variability in load-related responses between individuals, where some would experience considerable increases in postural angles, while others experienced considerable decreases.

Data related to key gait parameters, such as forefoot and heel forces, as well as gait speed, are shown in Figure 4. Significant load-related differences were evident for the forefoot force (M_diff_ = 34.86 N, *p* < 0.001), heel force (M_diff_ = 17.05 N, *p* < 0.001), and gait speed (M_diff_ = −0.18 km/hr, *p* = 0.016).

Changes in forefoot and heel forces are not surprising given that additional load was being carried. On closer inspection, the increases in forefoot and heel force were proportional to the increased load. Subsequently, we investigated the ‘ratio of forces’ (that is, forefoot force divided by heel force) to determine whether participants exhibited a greater forward shift in the center of mass post-loading compared to pre-loading. There were indeed significant increases in the force ratio from pre-loading (M = 1.38) compared to post-loading (M = 1.40; M_diff_ = 0.02, *p* = 0.029).

An unplanned follow-up analysis involved evaluating whether relative bag mass was associated with changes in postural angles. The results of this analysis can be seen in Figure 5, showing that the regression slope is not significantly different from zero and that all associations are negligible.

The results of the differences in the proportions of those with perceived pain who tend to carry loads above the 15% BM threshold compared to those below the threshold are shown in Figure 6A. There were substantially fewer individuals who reported pain when carrying loads below the 15% BM threshold (*n* = 3) compared to those above the threshold (*n* = 19). More specifically, the odds ratio was calculated to be 8.71 (95% CI (2.45, 41.73), *p* = 0.002), indicating that the odds of reporting pain when carrying loads above 15% BM were almost 9 times greater compared to those below the threshold. Differences with regards to the perceptions of pain between gender groups are shown in Figure 6B. In terms of relative proportions, there were no significant differences between genders for reporting pain (χ^2^_Pearson_ (1) = 0.75, *p* = 0.390).

The relative bag mass of the children from the present study were compared to similarly aged children from other countries to better contextualize the loads carried (see Figure 7). Children in the present study carried significantly heavier loads compared to most other counties (*p*_Holm_ < 0.001) with the exception of Italy (*p*_Holm_ = 0.694), Poland (*p*_Holm_ = 0.791), China (*p*_Holm_ = 0.791), and Malta (*p*_Holm_ = 0.791) (see Appendix A for a more detailed overview of the inferential statistics; and Appendix A for the publications used for this analysis).

## 4. Discussion

The novel findings of the present study indicate that acute loading (i) is different between age cohorts, whereby 13-year-olds tend to carry significantly lighter relative loads compared to younger cohorts, (ii) causes deviations in measured postures that are unlikely to be meaningful because of large between-subject variability, (iii) alters the relative proportion of forefoot-to-heel forces, implying a greater forward shift in the center of mass (CoM), (iv) tends to lead to significantly higher odds of reporting pain when loading exceeds 15% BM, and (v) tends to be higher in children from the current study when evaluated against similarly aged children across different countries.

Compared to the study by Puckree et al. [25] and Mwaka et al. [40], the participants in the present study were significantly lighter (M_diff_ = 4–8 kg, *p* < 0.001) and carried considerably heavier loads (M_diff_ = 3.79–4.01 kg, *p* < 0.001), although they were of similar mean age and sociodemographic background. We extended this further by showing that the loads carried in the present study were considerably heavier even when compared to international standards. A significant proportion of the available literature is based on data from the Northern Hemisphere (~88%), especially Europe (~48%), where sociodemographic and economic factors are considerably different to those in South Africa [34]. Therefore, substantially higher loads in the present study may be accounted for, at least partially, by factors such as access to lockers, security, quality of education, and quantity of homework [35,36,41]. The literature also showed that most schoolchildren within comparable age ranges tend to carry absolute loads of 5–7 kg [2] that fall within the acceptable 10–15% BM range. An intriguing finding of the present study was that 13-year-olds carried significantly lighter bags (~5.56 kg; 11%BM) compared to 10–12-year-old children (~6.87 kg; 18–20% BM). Such a finding might suggest that, as students continue to grow into puberty, the increases in bag mass tend to plateau, resulting in comparably lighter relative loads. Whether such a trend continues for older age cohorts would require further research.

The transition from the unloaded to loaded state caused acute changes in the static postural angles. Although the changes in all measured angles were not significantly different, it is important to note the within-subject variability caused by loading. Participants could experience increases or decreases in postural angles of more than 10–20° in some instances, but the direction of the change was arbitrary and independent of the load, as shown by the regression analysis. Although CHA, CVA, and SSP have been mentioned numerous times in the literature [1,8,24], it is likely that these angles lack the sensitivity and/or specificity to reliably measure changes in posture as a function of load or over time.

During dynamic tasks such as gait, significant elevations in forefoot and heel forces were observed, as well as reductions in gait speed because of loading. Higher foot forces are expected since a greater load is being carried with the increases being proportional to the load. For this reason, we investigated the force ratio (forefoot force divided by heel force) during the unloaded state, as this gives an indication of how much more force is being distributed on the forefoot relative to the heel and whether this ratio changed by a meaningful amount once loaded. Although significant changes in the force ratio were evident (M_diff_ = 0.02, *p* = 0.029; ~2%), this is unlikely to be meaningful based on the fact that an approximately 17% change in load only led to a 2% change in forward weight-shifting. However, this shift does imply that forefoot loading was amplified to some extent to compensate for a load-imposed modification in the CoM. Stated differently, it is likely that participants compensated for the additional load by adjusting their whole-body position to lean more forward rather than adjusting the neck and/or shoulder postures. Similarly, a significant decrease in mean gait speed was recorded as participants transitioned from unloaded to loaded conditions, but this appears to be independent of the loading classification. Our findings in this regard are on par with those of the literature in that loading led to slower gait speeds, but it is important to highlight the substantial inter-individual variability, as well as the fact that the change in gait speed appears to be independent of load, as shown by our regression analysis [2,5,42]. Some research has shown that under loaded conditions, the speed of walking is expected to decrease so that energy expenditure is minimized [43]. However, loads up to 20% BM have not induced significantly higher workloads in children compared to unloaded walking, especially when the carry duration was relatively short (<10 min) [44].

A notable agreement in the literature pertains to the tolerable upper ‘threshold’ for load carriage in younger age cohorts being ~10–15%BM [2,3,9,45]. The current study appears to agree on the basis that those carrying loads above 15% were ~9 times more likely to report pain associated with bag carrying compared to those below the threshold. A possible explanation for the higher carry loads may be the lack of access to lockers, greater amounts of homework, and larger textbooks [35,36,41]. It is noteworthy to mention that, despite at least two decades of research, there appears to be a lack of implementation of policies to mitigate excessive loading, especially in younger age cohorts. Based on the available evidence, although limited, schoolbag loads appear to have increased over time, coupled with the fact that levels of physical activity have declined appreciably, potentially putting pupils at greater risk of musculoskeletal injury [46,47]. We hope that the data from the present study may serve as motivation for schools and the Department of Basic Education to consider transitioning to electronic platforms for textbooks and homework as a potential means of minimizing additional load. The use of lockers and more ergonomically designed schoolbags could also serve as viable, potentially cost-effective alternatives in more resource-limited schools.

Future research should investigate whether acute interventions aimed at changes in physical activity and/or strength would alter perceptions of pain and the ability to tolerate higher loads may be worthwhile. Furthermore, longitudinal research should be conducted to trace whether more chronic changes occur in specific individuals across time.

Some limitations related to the present study must be noted. We did not control for differences in socioeconomic status, which is an important component in a South African context because of large variability in educational quality across socioeconomic domains. A greater range of ages should be explored to determine whether trends are maintained across older age cohorts. We were unable to account for potential confounders such as physical activity, strength, and sedentary time between individuals. Whether a more prolonged period of static standing (e.g., 5 min vs. 20 s) would lead to more noticeable changes in posture and/or gait would require further investigation and is noted as a limitation.

## 5. Conclusions

In conclusion, the present study is one of the first within the South African context to show that individual load carriage within the 10–13-year age range tends to substantially exceed the prescribed limit of 15% BM. It is probable that the demands of school may mandate such loads, but it should be noted that such carrying capacities are associated with load-related reports of pain. Changes in posture and gait parameters appear to be independent of load, and it is likely that measures of CVA, CHA, and SSP lack sensitivity or specificity to detect meaningful changes caused by large within-subject and between-subject variability. Although acute changes in gait parameters were detected, these are unlikely to be meaningful in the short term. Longer carrying durations should be investigated, as well as longitudinal follow-ups with the same participants throughout their schooling career to establish whether long-term effects are present.

## Figures and Tables

**Figure 1 children-10-01497-f001:**
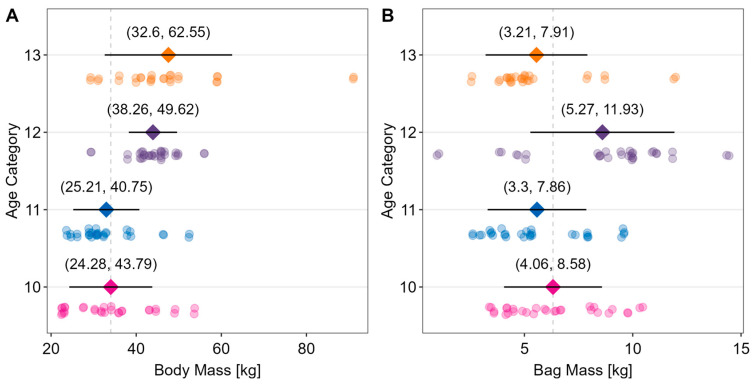
Descriptive statistics for body mass and bag mass across the different age ranges. Data are shown as mean ± SD. Vertical grey dashed line shows mass distributions relative to the 10-year-old age cohort. Panel (**A**) shows the individual data points and mean ± standard deviation (SD) of body mass of as a function of age. Panel (**B**) shows the individual data points and mean ± standard deviation (SD) of the relative bag mass (%BM) carried as a function of age.

**Figure 2 children-10-01497-f002:**
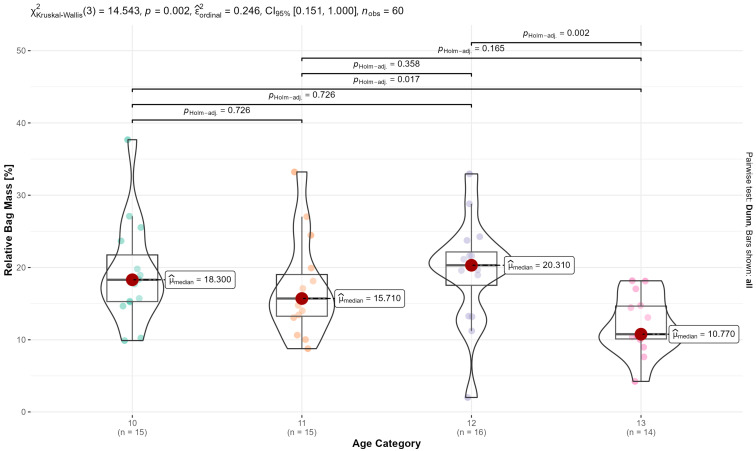
Inferential statistics showing mean differences in relative bag mass as a function of the age category.

**Figure 3 children-10-01497-f003:**
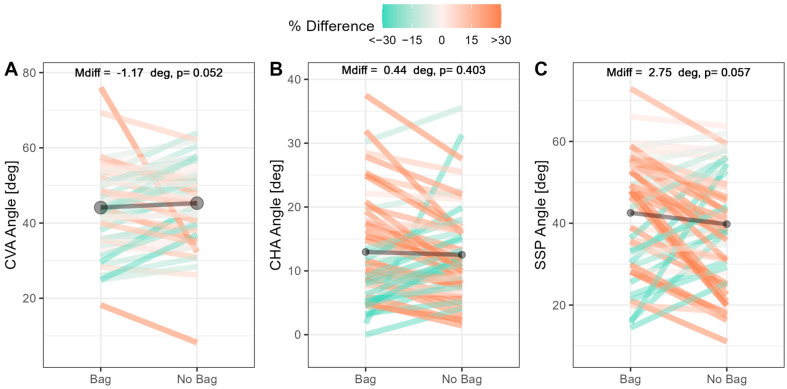
Inferential statistics showing differences in postural angles during loaded and unloaded conditions. Panels (**A**–**C**) show loaded vs. unloaded comparisons for CVA, CHA, and SSP; dark bars show the mean for each condition. Note: CHA = craniohorizontal angle, CVA = craniovertebral angle, and SSP = sagittal shoulder posture.

**Figure 4 children-10-01497-f004:**
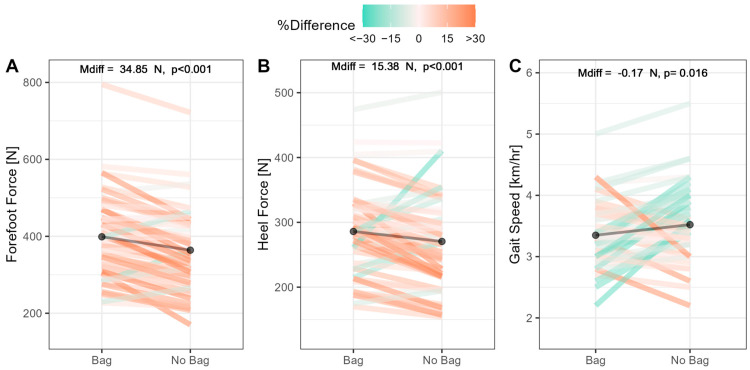
Inferential statistics showing differences in gait parameters during loaded and unloaded conditions. Panels (**A**–**C**) show loaded vs. unloaded comparisons for forefoot force, heel force, and gait speed; dark bars show the mean for each condition.

**Figure 5 children-10-01497-f005:**
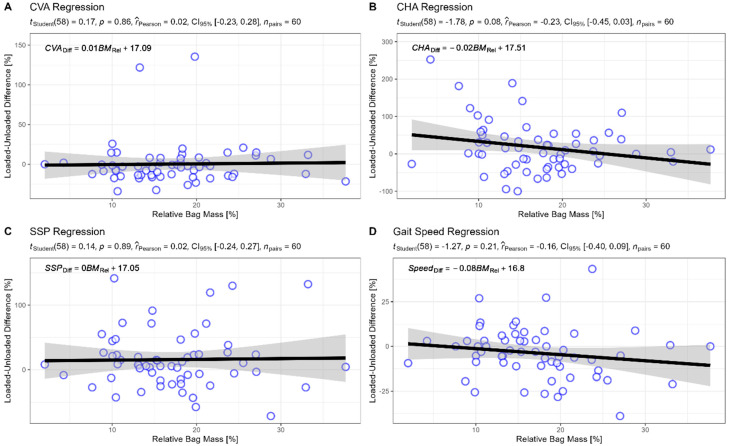
Regression analysis between relative bag mass and the associated changes in postural angles and gait speed. Note: CHA = craniohorizontal, CVA = craniovertebral, SSP = sagittal shoulder posture; black line indicates the regression line; grey area indicates the 95% CI of the regression line.

**Figure 6 children-10-01497-f006:**
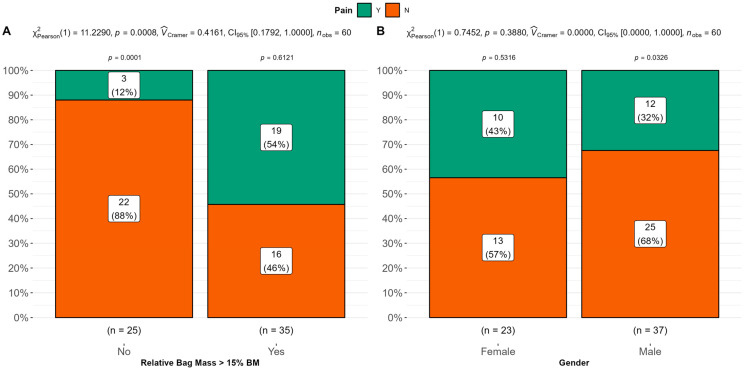
Inferential statistics showing proportional differences in the reporting of pain. Panel (**A**) shows differences between those who experience pain as a function of ‘excessive’ loading (>15% BM) compared to those carrying loads below 15% BM. Panel (**B**) shows differences in between genders in the reporting of pain.

**Figure 7 children-10-01497-f007:**
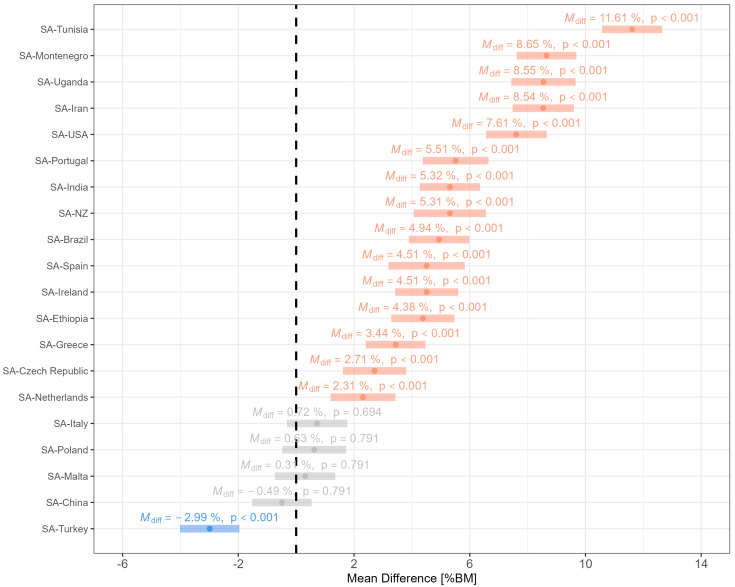
Relative bag mass of children from South Africa in comparison to those from other countries. Note: red = relative bag mass from SA is significantly greater than that from other countries; grey = similar relative bag mass between countries; blue = relative bag mass from SA is significantly less than that from other countries. The vertical black line indicates zero difference between countries. For more details about the inferential statistics see Appendix A. The data and publications used for this figure are available in Appendix A.

## Data Availability

The dataset for the study is freely available at the Harvard Dataverse and can be accessed via the following link: https://doi.org/10.7910/DVN/HXRMMI.

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
