# Peer review of "The Acute Effects of Schoolbag Loading on Posture and Gait Mechanics in 10- to 13-Year-Old Children: A Cohort from the North West Province"

_children, 2023, doi:10.3390/children10091497_

Round 1

Reviewer 1 Report

I would like to congratulate the authors who have presented this study. Undoubtedly, it is a research that refers to a very interesting subject of study in the educational field. 

The research has been developed in an effective way and following the steps of the scientific method, however, I think it needs some improvement. 

In the results section I suggest adding tables to better understand the results. 

Add future perspectives that may arise from this study. Also add areas of application.

Author Response

Response to Reviewers

Reviewer 1

Comment 1: I would like to congratulate the authors who have presented this study. Undoubtedly, it is a research that refers to a very interesting subject of study in the educational field. 

Response 1: Thank you very much.

Comment 2: The research has been developed in an effective way and following the steps of the scientific method, however, I think it needs some improvement. 

Response 2: Thank you, we will try to incorporate the suggested changes to improve the overall quality of the manuscript.

Comment 3: In the results section I suggest adding tables to better understand the results. 

Response 3: Thank you for the suggestion. We have provided more statistical details for the relevant figures (especially Figure 7) and we have included a supplementary table (S1) related to Figure 7 for those readers that would like additional information (Mdiff  ± 95% CI, margin of error, p-value). The data and publications used for the data are available in a separate supplementary file (S2).

Comment 4: Add future perspectives that may arise from this study. Also add areas of application.

Response 4: Thank you for the suggestion. We have added more detail in the discussion section of the manuscript.

Reviewer 2 Report

Dear Authors,

Thank you for the opportunity to review this manuscript. The research results address the important issue of biological consequences, both short-term and long-term, of too heavy school bags in a group of children in adolescence. In my opinion, the manuscript requires major correction, especially in the methodological and therefore basic area. 

Comments and suggestions for authors:

1. Please explain why the results of the research on children in South Africa may be different from similar results available in other populations? Please explain in the Introduction section.

2. The question of age - How exactly are the age groups separated? For example, among 10-year-olds, are there children of calendar age 9.5-10.5 or 10.0-10.99? Please explain in detail in the methodological section.

3. The issue of gender - In my opinion, a significant and fundamental methodological error is the failure to take into account the gender of children in the division. Girls and boys aged 10-13 differ diametrically in their biological age and sexual maturity. Girls mature two years faster on average. It is not allowed to provide values of somatic parameters (e.g. height and weight averaged for both sexes).

4. The issue of puberty - Lack of information about the biological age of the respondents and the stage of maturity. Children between the ages of 10 and 13 can vary greatly in their sexual maturity. The difference in biological status and sexual maturity between early and late maturing children can be as much as 6 calendar years. This is a very strong limitation of this analysis. Thus, in the group of e.g. age, e.g. 16 years old. Please refer to this issue.

5. In the scientific text, the results in tabular form are the basis, and the figures are supplementary. The results shown only in the form of figures do not provide the possibility of citing exact results. Please present the basic results in the form of clear tables. You may consider including additional materials.

6. Figure 7 should be in the Discussion section, in my opinion.

Sincerely,

rewevier

Author Response

Response to Reviewers

Comments and suggestions for authors:

  1. Please explain why the results of the research on children in South Africa may be different from similar results available in other populations? Please explain in the Introduction section.

Response: Thank you for the comment and suggestion. We have subsequently added more detail within the introduction section to better contextualise the importance of the research within a South African context and why it may be different when compared to other countries.

  1. The question of age - How exactly are the age groups separated? For example, among 10-year-olds, are there children of calendar age 9.5-10.5 or 10.0-10.99? Please explain in detail in the methodological section.

Response: All children within the 10-year-old age range would be between 10.0-10.99. We have added this information in the relevant section of the manuscript.

  1. The issue of gender - In my opinion, a significant and fundamental methodological error is the failure to take into account the gender of children in the division. Girls and boys aged 10-13 differ diametrically in their biological age and sexual maturity. Girls mature two years faster on average. It is not allowed to provide values of somatic parameters (e.g. height and weight averaged for both sexes).

Response: Thank you for the comment. We have added information that takes into account the gender of the children for each age cohort. It is important to note that the purpose of the study was not to evaluate gender-specific differences, but to focus rather on specific age cohorts since the school curriculum would prescribe textbooks, homework, and schoolwork based on age/class and not gender. It is therefore important to report differences in loading parameters as a function of age. More specifically, evidence shows that no significant differences (p>0.11) are present in terms of strength for males (11.8 ± 0.4 years) and females (11.9 ± 0.4 years) and should therefore not affect the ability to tolerate the loads carried (DOI: 10.1055/s-2007-971955). We did mention this as a limitation whereby factors such as strength were not evaluated in the current study. However, we have subsequently added the statistical evaluation related to whether significant differences were present in terms of reporting pain between the gender groups (see Figure 7B).

  1. The issue of puberty - Lack of information about the biological age of the respondents and the stage of maturity. Children between the ages of 10 and 13 can vary greatly in their sexual maturity. The difference in biological status and sexual maturity between early and late maturing children can be as much as 6 calendar years. This is a very strong limitation of this analysis. Thus, in the group of e.g. age, e.g. 16 years old. Please refer to this issue.

Response: Thank you. While this is true there was no evidence for differences in body mass between genders in the same age cohort. Based on some available literature, substantial differences are likely to be expected after the age of 12-13 years. We have subsequently added the statistical evidence for the lack of differences in body mass between genders in each age cohort. We have also included a supplementary table that provides greater insights in terms of statistical inferences. Furthermore, we had indicated previously that we did not evaluate differences in strength, and that this was considered a limitation of the study.

  1. In the scientific text, the results in tabular form are the basis, and the figures are supplementary. The results shown only in the form of figures do not provide the possibility of citing exact results. Please present the basic results in the form of clear tables. You may consider including additional materials.

Response: Thank you for the comment and suggestion. Determining the best way to present the data is always a difficult part of any manuscript. Providing information in a table has major limitations in that individual data points of participants cannot be shown; instead, only summary statistics can be shown. We have therefore opted for the following details for each figure which cannot be communicated in a table format:

Figure 1 shows the mean ± SD together with individual data points for each age cohort with appropriate ranges for precision.

Figure 2  shows individual data points, median and interquartile range for each age cohort. Inferential statistics for the ANOVA as well as paired comparison are shown according to the latest statistical conventions.

Figure 3 shows individual data points for each of the measured postural angles for both the bag and no-bag conditions. More specifically the figure shows the differences in percentage terms based on the color scale  where green implies a decrease and orange implies an increase. The intensity of the color indicates the magnitude of the change where darker color gradients reveal larger changes, and lighter color gradients imply smaller changes. The mean change is also shown, as well as the inferential statistics associated with the change. The amount of information shown in just this figure cannot be replicated in a table, and we therefore would like to retain this figure.

Figure 4 shows the information in the same format, with the same level of detail as we have stipulated for Figure 3

Figure 5 shows the individual data points, the regression line and confidence interval for the regression line. We have also shown that inferential statistics for each regression line according to the latest statistical conventions. Again, the amount of information conveyed in the figure cannot be conveyed in tabular format and we would therefore like to keep this figure in its present form.

Figure 6 highlights the differences in proportions between those individuals that reported pain and those who did not, based on the loads carried. We have opted to include an analysis based on gender differences whereby we show whether differences were present in the proportion of males vs females that reported pain. Again, all the results report the full suite of statistical details based on the most recent conventions for accurate and complete statistical reporting standards.

Figure 7 shows the differences in mean relative bag mass carried by children across different countries. Here we agree that more information could be provided to make the differences more clear and precise for reporting purposes. Furthermore, we have also included a table with the relevant details in a supplementary file so that readers can review more precise information should the wish to do so.

  1. Figure 7 should be in the Discussion section, in my opinion.

Response: Thank you. We respectfully disagree on the basis that this figure forms part of an original analysis which presently does not exist and therefore should be part of the results section. In other words, our research team completed a detailed statistical analysis that incorporates both summary and inferential statistical techniques to determine whether differences in relative schoolbag mass were present between South African children and children from other countries. The results are therefore novel to the present study and should therefore be part of the results section.  

Reviewer 3 Report

Dear All,

I have read his work with great interest. It seems good to me. I only leave some recommendations that aim to improve what they already have.

I hope you find it useful.

Line 132, it should say PRISMA

The articles used to compare should be shown and briefly indicate the characteristics of each one. They could be attached as supplementary material.

It could be a limitation of your study not to analyze how your posture is throughout the day, regardless of whether or not you wear a backpack

I suggest incorporating the strengths of your research, as well as the usefulness of your results for practice.

Author Response

Response to Reviewers

Comment: Line 132, it should say PRISMA

Response: Thank you, this was a typo from our side and we have corrected it. 

Comment: The articles used to compare should be shown and briefly indicate the characteristics of each one. They could be attached as supplementary material.

Response: Thank you. We agree and have provided a supplementary file that highlights the literature that was sourced to provide the information used to create the figure. We have also provided additional statistical information in the figure itself.

Comment: It could be a limitation of your study not to analyze how your posture is throughout the day, regardless of whether or not you wear a backpack

Response: Thank you. I believe that we did mention this is a limitation (e.g. whether differences were present if pupils carried bags for substantially longer periods). We have also included a statement that longitudinal research should be conducted on the same individuals as they continue to age, to determine whether changes tend to occur across time.

Comment: I suggest incorporating the strengths of your research, as well as the usefulness of your results for practice.

Response: Thank you for this suggestion. We have subsequently added elements of this in the discussion.

Round 2

Reviewer 1 Report

Thank you very much for adjusting the article to the requested changes. I believe that the article has been significantly improved and I authorise its publication. 

Reviewer 2 Report

Dear Authors,

Many thanks to the Authors for responding to my comments. I accept information from the Authors, thank you for the appropriate additions to the text. In the context of my comments, I believe that it is very important to provide the exact limitation study. I do not agree with the authors' statement that their goal was not to study dimorphic differences, so they do not provide values separately for boys and girls. As a human biologist, I firmly state that the results of, for example, biological characteristics for both sexes should not be averaged. There are simply no such average individuals. And giving the values of traits for boys and girls is simply scientific reliability, and is not always intended to, for example, analyze dimorphic differences. I am aware that some publications present average values, but I believe that they are incorrect. I leave the decision to the Editors.

Kind regards,

revewier